# Synthesis and Modeling Studies of Furoxan Coupled Spiro-Isoquinolino Piperidine Derivatives as NO Releasing PDE 5 Inhibitors

**DOI:** 10.3390/biomedicines8050121

**Published:** 2020-05-14

**Authors:** Swami Prabhuling, Yasinalli Tamboli, Prafulla B. Choudhari, Manish S. Bhatia, Tapan Kumar Mohanta, Ahmed Al-Harrasi, Zubaidha K. Pudukulathan

**Affiliations:** 1School of Chemical Sciences, SRTM University, Nanded 431606, Maharashtra, India; sprabhuling@gmail.com (S.P.); yasinmedchem@gmail.com (Y.T.); 2Department of Pharmaceutical Chemistry, Bharati Vidyapeeth College of Pharmacy, Kolhapur 416013, Maharashtra, India; praffula12@rediffmail.com (P.B.C.); bhatiamanish13@gmail.com (M.S.B.); 3Natural and Medical Sciences Research Center, University of Nizwa, P.O. Box 33, Postal Code 616, Birkat Al Mouz, Nizwa, Oman

**Keywords:** nitric oxide, furoxan, PDE 5 inhibitors, spiro-isoquinolino piperidine

## Abstract

Nitric oxide (NO) is considered to be one of the most important intracellular messengers that play an active role as neurotransmitter in regulation of various cardiovascular physiological and pathological processes. Nitric oxide (NO) is a major factor in penile erectile function. NO exerts a relaxing action on corpus cavernosum and penile arteries by activating smooth muscle soluble guanylate cyclase and increasing the intracellular concentration of cyclic guanosine monophosphate (cGMP). Phophodiesterase (PDE) inhibitors have potential therapeutic applications. NO hybridization has been found to improve and extend the pharmacological properties of the parental compound. The present study describes the synthesis of novel furoxan coupled spiro-isoquinolino-piperidine derivatives and their smooth muscle relaxant activity. The study reveals that, particularly **10d** (1.50 ± 0.6) and **10g** (1.65 ± 0.7) are moderate PDE 5 inhibitors as compared to Sidenafil (1.43 ± 0.5). The observed effect was explained by molecular modelling studies on phosphodiesterase.

## 1. Introduction

Approximately 50% of men aged over 40 years suffer from male erectile dysfunction. The relaxation of arterial and trabecular smooth muscle is needed to achieve and maintain the penile erection where vascular diseases have impaired the vasodilator responses. This vasodilator response has been associated with the development of erectile dysfunction and impotence. However, the treatment options have widened since the launch of the phosphodiesterase type 5 (PDE5) inhibitor, sildenafil citrate (Viagra™). The absence of endogenous nitric oxide (NO) could be rectified by the potential use of NO releasing PDE5 inhibitors under NO-deficient conditions [1].

NO is commonly known as one of the important intracellular messengers and a neurotransmitter [2]. It plays an important role in regulation of various cardiovascular, central nervous system, and immune system physiological and pathological processes [3,4,5]. The NO is released by the endothelial lining of blood vessels in response to agonists such as acetyl choline [6]. Bradykinin has a relaxing effect [7,8] and continuous basal generation of NO prevents platelet aggregation [8], muscle cell mitogenesis [9], effects as a blocker of calcium channels, and attributes to analgesic properties [10,11]. This key endogenous factor plays a prominent role in providing protection against atherosclerotic and thrombotic processes that underpin development of various cardiovascular diseases. In the central nervous system, NO is an important neurotransmitter and neuromodulator [12] and in the immune system, inflammatory cells produce a high concentration of NO in response to invading pathogens [13]. Nitric oxide (NO) is a key mediator of penile smooth muscle relaxation and is released by nonadrenergic, noncholinergic nerves within the trabecular and penile arterial tissues as well as by the endothelia that line the lacunar spaces and the intima of penile arteries [14,15,16,17]. NO exerts its relaxing action on corpus cavernosum and penile arteries by activating smooth muscle soluble guanylate cyclase and increasing the intracellular concentration of cGMP [18,19,20,21].

Phosphodiesterase (PDE) are enzymes which regulate the cellular concentration of cAMP (cyclic adenosine monophosphate) and cGMP [22,23]. In humans, there are 21 PDE genes which are classified into the 11 families [24]. Due to their involvement in various important biological functions the PDE has emerged as a promising target for the treatment of various human diseases, particularly cardiovascular disease [25,26,27,28,29]. The PDE 5 enzyme is abundantly expressed in smooth muscle cells [30] of the penis [31], lungs [32], kidney [33], spleen, as well as platelets [34], all of which rely heavily on the cyclic guanosine monophosphate (cGMP) signaling cascade. Marketed inhibitors of the enzyme, such as sildenafil and vardenafil, are capable of producing acute vasodilatory effects through inhibition of cGMP hydrolysis. The cGMP accumulation augments the nitric oxide (NO) induced vascular smooth muscle relaxation [35,36].

Thus, the identification of more potent and more selective PDE 5 inhibitors is of substantial medicinal and commercial interest [37,38,39]. The PDE 5 inhibitors have therapeutic potential for the treatment of heart diseases as they act as a vascular smooth muscle relaxant and bring vasodilation. The standard drug sildenafil is a well-known PDE5 inhibitor used to affect vasodilation and treatment of various CVS disorders. The developments of new therapeutically active PDE 5 inhibitors continue to attract the attention of synthetic and medicinal chemists. The spirocycles with distinct structural features are important scaffolds often found in natural products and display numerous bioactivities. In particular, spiropiperidines are considered as privileged structures and have received considerable attention in medicinal chemistry as the skeleton displays a wide array of activities [40,41]. 

The NO–cGMP mediated relaxation pathway has been elucidated earlier whereas the smooth muscle tissues differ in their sensitivity to this pathway. The molecular basis of the smooth muscle functional diversity is not well understood. However, the role of NO is well defined in erectile dysfunction but the effect of exogenous NO donor particularly Furoxan or NO donor hybrid is not explored.

NO hybridization has been found to improve the pharmacological properties of the parental compound. A considerable number of NO hybrid compounds have been synthesized and screened. The role of NO donor hybrid compounds has been extended from the initial inflammatory and cardiovascular diseases to gastrointestinal disorders, microbial infections, cancer, neurodegenerative diseases, sexual dysfunction, ocular hypertension, and others [42,43]. NO-hybridization of parental drugs such as aspirin [44], isoxazoline compounds (VGX-1027), and antiretroviral protease inhibitors has been largely studied and NO hybridization has been found to often improve and eventually extend the pharmacological properties of parental compounds in suppression of cancer and immunoinflammatory responses [45,46]. 

Along with NO, H_2_S and carbon monoxide represent the endogenous gaseous system that is implicated in control of several biological responses [47]. It is important to notice that some biological effects of NO and H_2_S are synergistic, including the effects mediated by PDE4 inhibition on relaxation of pig and human [48].

Keeping in view the potential application of PDE 5 inhibitors, development of new novel NO releasing donors of medicinal significance continues to attract the attention of researchers. The present work focuses on exogenous supply of NO from hybrid donors coupled to PDE 5 inhibitors. Thus, simultaneous release of NO and inhibition of PDE 5 has been explored to potentiate the desired effect. In the present study, furoxan coupled spiro-isoquinolino piperidine skeleton has been synthesized and explored as an effective PDE 5 inhibitors.

## 2. Material and Methods

All the regents and solvents used are commercially available. The compounds were purified by column chromatography technique using silica gel (60–120 mesh). The melting points were determined using the open capillary method and are uncorrected. ^1^H NMR and ^13^C NMR spectra were recorded on either a Brucker Avance 300 MHz or a Varian Inova 400 and 500 MHz FT spectrometer using TMS as an internal standard (chemical shift in d values, *J* in Hz). The high-resolution mass spectra and electrospray ionization recorded on QSTAR XL high resolution mass spectrometer. 

**Synthesis of N-benzyl-2-chloro-N-(2-chloroethyl) ethanamine (2):** To a stirred suspension of compound bis(2-chloroethyl) amine hydrochloride (10 mmol) and K_2_CO_3_ (25 mmol) in ACN (10 mL), benzyl bromide (10 mmol) was added dropwise and reaction mixture was stirred for 24 h at ambient temperature. After completion of the reaction, the reaction mass was quenched in cold water and extracted with ethyl acetate. The organic layer was extracted with aqueous HCl. The aqueous layer basified with saturated K_2_CO_3_ and extracted with ethyl acetate. The organic layer was washed with brine solution, dried over anhydrous sodium sulphate, and concentrated under reduced pressure to afford the pure compound **2** (yield: 73%) as pale-yellow liquid. This was used in the next step without isolation.

**Synthesis of 1-benzyl-4-phenylpiperidine-4-carbonitrile (3):** To a cold stirred suspension of NaH (20 mmol) in DMF (60 mL), 2-phenylacetonitrile (10 mmol) was added dropwise at 0–5 °C and stirred for 30 min at the same temperature. To this, a solution of compound **2** (10 mmol) in DMF (15 ml) was added dropwise at 0 °C. The resulting reaction mixture was heated at 60 °C for 6 h. After the completion of the reaction, the reaction mass was quenched in ice water and extracted with ethyl acetate. The organic layer was washed with brine, dried over anhydrous sodium sulphate, and concentrated under the reduced pressure to afford the crude compound **3** as a brown solid with 75% yield. This was used in the next step without isolation.

**Synthesis of 1-benzyl-4-phenylpiperidine-4-carboxylic acid (4):** To a stirred solution of compound **3** (10 mmol) in ethanol (25 mL), 2N NaOH (15 mmol) was added and refluxed for 7 h. After the completion of reaction, the reaction mass was allowed to cool to room temperature, and concentrated under the reduced pressure to get the crude compound. The crude compound was dissolved in water and acidified to pH ≈5 using diluted HCl and stirred for 30 min. The obtained solid was filtered, washed with water, followed by acetone, and dried under the vacuum at 60 °C to afford the pure compound **4** as an off-white solid with 70% yield. The ESI-MS (*m/z*): 296 [M + H].

**Synthesis of 1-benzyl-4-phenylpiperidine-4-carbonyl chloride (5):** A stirred mixture of compound **4** (10 mmol), thionyl chloride (5 mL) in DMF (0.2 mL) refluxed for 10 h. After the completion of the reaction, the reaction mass was concentrated under reduced pressure to afford compound **5** with 87% yield. The compound **5** was used in the next step of the reaction without isolation.

**Synthesis of methyl 2-(1-benzyl-4-phenylpiperidine-4-carboxamido)acetate (6)**: To a cold solution of methyl 2-aminoacetate (22 mmol) and triethylamine (15 mmol) in DCM (10 mL), compound **5** (10 mmol) was added portion-wise at 0 °C and stirred for 12 h at room temperature. After the completion of the reaction, the reaction mass was diluted with water and extracted with DCM. The organic layer was washed with aqueous sodium bicarbonate solution followed by brine, dried over anhydrous sodium sulfate, and concentrated under the reduced pressure to afford the crude compound **6**. The crude compound **6** was triturated with hexane to afford the pure compound **6**, as an off-white solid, with yield of 83.0%. The ESI-MS (*m/z*): 367 [M + H].

**Synthesis of 3-benzyl-((3-oxo-2,3-dihydro-1H-spiro[isoquinoline-4,4']-piperidine (7):** A solution of compound **6** (10 mmol,) in trifluoromethane sulfonic acid (15 mmol,) was heated to 130 °C for 4 h. After the completion of the reaction, the reaction mass was quenched with chilled 50% NaOH solution and extracted with ethyl acetate. The organic layer was washed with brine solution, dried over anhydrous sodium sulphate, and concentrated under reduced pressure to afford the crude compound **7**. The crude compound **7** was triturated with hexane to afford the pure compound **7**, as an off-white solid, with yield of 58%. The compound **7** was used in the next step of the reaction without isolation.

**Synthesis of 3-tert-butyloxycarbonyl-((3-oxo-2,3-dihydro-1H-spiro[isoquinoline-4,4'] -piperidine (8):** Under nitrogen atmosphere to a solution of compound **7** (10 mmol), Boc anhydride (12 mmol) and triethylamine (20 mmol) in methanol (10 mL), 10% Pd/C (10 mol %) was added. The resulting mixture was hydrogenated under 40 psi hydrogen pressure at room temperature for 16 h. After the completion of the reaction, the reaction mass was filtered through celite and washed with methanol. The filtrate was concentrated under reduced pressure to afford the crude compound **8**, which was triturated with hexane to afford the pure compound **8** as an off-white solid with 55% yield. ^1^H NMR (400 MHz) DMSО-D_6_: δ 8.18 (1H, br NH), 7.29–7.42 (1H, m), 7.19–7.32 (3H, m), 4.41 (2H s), 3.69-3.79 (2H, m), 3.22–3.41 (2H m), 1.91–2.21 (m, 2H) 1.62–1.78 (2H, m), 1.41 (9H s).

**Synthesis of 3-(3-oxo-2,3-dihydro-1H-spiro[isoquinoline-4,4']-piperidine. hydrochloride (9):** A solution of compound **8** (10 mmol) in dioxane. HCl (10 mL) was stirred at room temperature for 12 h. After completion of reaction, the reaction mass was concentrated under reduced pressure, triturated with hexane to afford pure compound **9**, as an off-white solid with 74.4% yield. The compound **9** was used in the next step of the reaction without isolation.

**Synthesis of compounds 10a–j**: To the stirred solution of compound **9** (10 mmol) in DMF (5 mL), TEA (20 mmol) was added at 0–5 °C and stirred for 30 min. Bromo furoxan (10 mmol) was added portion-wise and stirred at room temperature for 4 h. After completion of the reaction, the reaction mass was evaporated under reduced pressure and the reaction mass was quenched with cold water and extracted with ethyl acetate. Then, the organic layer was concentrated and the residue was purified by column chromatography to afford compounds **10a–j** with yield of 50–65%. 

**3-((3-oxo-2,3-dihydro-^1^H-spiro[isoquinoline-4,4'-piperidin]-1'-yl)methyl)-4-phenyl-1,2,5-oxadiazole 2-oxide** (**10a**): Off white solid, 168–171 °C. ^1^H NMR (400 MHz) CDCl_3_: δ 8.00–8.02 (2H, d, *J = 8 Hz*), 7.52–7.54 (3H, d, *J = 8 Hz*), 7.43–7.45 (1H, d, *J = 8 Hz*), 7.30–7.34 (1H, m), 6.43 (1H, br, NH), 4.49 (2H, s), 3.60 (2H s), 2.74–3.00 (4H, m), 1.96–2.22 (4H, m). ^13^C NMR (100 MHz) CDCl_3_: 175.92, 157.65, 140.69, 132.74, 131.02, 129.08, 127.75, 126.97, 126.63, 125.95, 124.55, 113.05, 50.61, 45.09, 43.32, 31.77. HRMS: 391.1803 whereas calculated mass: 391.1783.

**4-(4-methoxyphenyl)-3-((3-oxo-2,3-dihydro-1H-spiro[isoquinoline-4,4'-piperidin]-1'-yl) methyl)-1,2,5-oxadiazole 2-oxide(10b):** Pale yellow solid, 176–178 °C. ^1^H NMR (300 MHz) CDCl3: δ 7.95–7.97 (2H, d, *J = 6 Hz*), 7.30–7.46 (2H, m), 7.15–7.24 (2H, m), 7.01–7.03 (2H d, *J* = 6 Hz) 6.20 (1H, br, NH) 4.49–(2H, s), 3.88 (3H s), 3.58(2H, s) 2.74–3.00 (4H, m) 1.97–2.23 (4H, m). ^13^C NMR (75 MHz) CDCl_3_: 175.90, 157.16, 132.84, 129.70, 127.83, 126.68, 126.01, 124.62, 114.54, 96.21, 55.54, 96.21, 55.32, 50.70, 50.38, 45.15, 43.34, 31.89. HRMS: 421.18536 whereas calculated mass: 421.18703.

**4-(4-chlorophenyl)-3-((3-oxo-2,3-dihydro-1H-spiro[isoquinoline-4,4'-piperidin]-1'-yl) methyl)-1,2,5-oxadiazole 2-oxide(10c):** Pale yellow solid, 172–175 °C. ^1^H NMR (300 MHz) CDCl3: δ 7.99–8.02 ( 2H, d, *J = 9 Hz*), 7.49–7.52 (2H, d, *J = 9 Hz*) 7.42–7.45 (1H, d, *J = 9Hz*), 7.27–7.23 (2H, m) 6.39 (1H, br), 4.48 (2H, s), 3.58 (2H, s), 2.71–3.02 (4H, m), 2.01–2.23 (2H, m). ^13^C NMR (75 MHz) CDCl3: 174.06, 155.52, 139.89, 136.02, 132.44, 129.86, 126.44, 125.35, 124.88, 123.64, 111.47, 94.96, 66.66, 49.53, 49.18, 43.51, 41.93, 30.55. ESI-MS (*m/z*) 425 [M + H]+.

**4-(2-nitrophenyl)-3-((3-oxo-2,3-dihydro-1H-spiro[isoquinoline-4,4'-piperidin]-1'-yl) methyl) -1,2,5 -oxadiazole 2-oxide(10d):** Yellow solid, 185–189 °C. ^1^H NMR (300 MHz) CDCl_3_: δ 8.25–8.26 (1H, m), 7.71–7.74 (3H, m) 7.27–7.53 (2H, m) 7.10–7.21 (2H, m) 6.50 (1H, br), 4.39 (2H, s), 3.47 (2H, s), 2.35–2.65 (4H, m), 1.25–1.78 (4H, m). ^13^C NMR (75 MHz) CDCl_3_: 175.75, 155.57, 148.85, 140.64, 133.68, 132.08, 131.49, 127.64, 126.60, 125.91, 124.45, 124.45, 124.02, 113.67, 96.13, 52.13, 50.59, 44.88, 42.66, 31.23. HRMS: 436.16023 whereas calculated mass: 436.16155.

**4-(4-nitrophenyl)-3-((3-oxo-2,3-dihydro-1H-spiro[isoquinoline-4,4'-piperidin]-1'-yl)methyl)-1,2,5-oxadiazole 2-oxide(10e):** Yellow solid, 191–194 °C. ^1^H NMR (400 MHz) CDCl_3_: δ 8.37–8.39 (2H, d, *J* = *8 Hz*), 8.29-8.31 (2H, d, *J* = *8 Hz*) 7.43–7.41 (1 H, m)*,* 7.34–7.30 (1H, m), 7.23–7.27 (1H, m), 7.15–7.18 (1H, m), 6.25 (1H, br), 4.49 (2H, s), 3.61 (2H, s), 2.73–2.98 (4H, m), 1.63–2.23 (4H, m). ^13^C NMR (100 MHz) CDCl_3_: 175.65, 155.80, 149.24, 140.49, 132.61, 129.27, 127.90, 126.78, 126.02, 124.38, 124.22, 112.22, 50.61, 50.34, 45.11, 43.22, 31.67. HRMS: 436.16023 whereas calculated mass: 436.16155.

**4-(4-bromophenyl)-3-((3-oxo-2,3-dihydro-1H-spiro[isoquinoline-4,4'-piperidin]-1'-yl) methyl) -1,2,5-oxadiazole 2-oxide(10f):** Pale yellow solid, 174–176 °C. ^1^H NMR (400 MHz) CDCl_3_: δ 7.76–7.78 (2H, d, *J* = *8 Hz*), 7.48–7.51 (2H, d, *J* = 8 Hz), 7.43–7.46 (1H, m), 7.28–7.32 (2H, m), 7.18–7.21 (1H, m), 6.37 (1H, br), 4.56 (2H, s), 3.55 (2H, s), 2.71–2.89 (4H, m), 1.98–2.04 (4H, m). ^13^CNMR (100 MHz) CDCl_3_: 173.16, 155.02, 138.88, 135.93, 132.06, 129.56, 126.44, 123.04, 125.35, 124.88, 111.18, 94.75, 66.53, 48.93, 48.67, 43.41, 41.39, 30.02. ESI-MS (*m/z*) 470 [M + H]+.

**4-(2-methoxyphenyl)-3-((3-oxo-2,3-dihydro-1H-spiro[isoquinoline-4,4'-piperidin]-1'-yl) methyl)-1,2,5-oxadiazole 2-oxide (10g):** Pale yellow solid, 173–175 °C. ^1^H NMR (300 MHz) CDCl_3_: δ 8.37–8.39 (1H, m), 7.88-7.93 (3H, m), 7.38–7.63 (2H, m), 7.18–7.21 (2H, m) 6.4 (1H, br), 4.48 (2H, s), 3.79 (2H, s), 2.41–2.70 (4H, m), 1.26–1.82 (4H, m). ^13^C NMR (75 MHz) CDCl_3_: 178.70, 156.20, 148.97, 141.97, 135.86, 132.97, 131.97, 128.46, 126.64, 125.98, 125.89, 124.44, 113.97, 96.96, 55.94, 50.69, 46.23,43.39, 31.78. HRMS: 421.18536 whereas calculated mass: 421.18703.

**4-(3-nitrophenyl)-3-((3-oxo-2,3-dihydro-1H-spiro[isoquinoline-4,4'-piperidin]-1'-yl) methyl) 1,2,5-oxadiazole 2-oxide(10h):** Pale yellow solid, 183–185 °C. ^1^H NMR (300 MHz) CDCl_3_: δ 8.52–8.56 (1H, m), 8.16–8.21 (3H, m), 7.59–7.67 (2H, m) 7.29–7.49 (2H, m) 6.61 (1H, br), 4.45 (2H, s), 3.61 (2H, s), 2.45–2.67 (4H, m), 1.17–1.79 (4H, m). ^13^C NMR (75 MHz) CDCl_3_: 174.87, 155.93, 148.97, 140.74, 132.97, 132.74, 131.96, 127.90, 126.68, 125.67, 125.98, 124.32, 113.76, 96.74, 54.94, 50.91, 44.7, 32.97. HRMS: 436.16023 whereas calculated mass: 436.16155.

**4-(3-methoxyphenyl)-3-((3-oxo-2,3-dihydro-1H-spiro[isoquinoline-4,4'-piperidin]-1'-yl) methyl) -1,2,5-oxadiazole 2-oxide(10i):** Pale yellow solid, 176–179 °C. ^1^H NMR (300 MHz) CDCl_3_: δ 7.52–7.94 (4H, m), 7.24–7.49 (2H, m), 6.96–7.10 (2H, m)*,* 6.4 (1H, br), 4.46 (2H, s), 3.79 (2H, s), 2.32–2.47 (4H, m), 1.09–1.69 (4H, m). ^13^CNMR (75 MHz) CDCl_3_: 172.62, 160.21, 145.26, 142.34, 139.78, 134.21, 130.29, 125.90, 125.74, 118.92, 114.49, 111.94, 555.85, 54.32, 50.94, 44.74, 32.94. HRMS: 421.18536 whereas calculated mass: 421.18703.

**4-(3-bromophenyl)-3-((3-oxo-2,3-dihydro-1H-spiro[isoquinoline-4,4'-piperidin]-1'-yl) methyl)-1,2,5-oxadiazole 2-oxide(10j):** Pale yellow solid, 171–175 °C. ^1^H NMR (300 MHz) CDCl_3_: δ 7.91–7.93 (1H, m), 7.63–7.83 (3H, m) 7.38–7.58 (2H, m)*,* 7.10–7.27 (2H, m)*,* 6.4 (1H, br), 4.42 (2H s), 3.59 (2H, s), 2.35–2.52 (4H, m), 1.21–1.89 (4H, m). ^13^C NMR (75 MHz) CDCl_3_: 172.47, 149.27, 145.27, 139.97, 135.72, 131.41, 126.51, 126.92, 125.83, 125.43, 122.43, 111.93, 55.94, 50.97, 444.74, 32.96. ESI-MS (*m/z*) 470 [M + H]+.

### 2.1. Nitric Oxide Releasing Griess Assay: Quantitative Nitrite Detection

The nitric oxide release assay of the synthesized compounds was determined using the Griess assay as reported earlier [49]. A solution of compound (20 µL) in DMSO was added to 2 mL of 50 mM phosphate buffer (pH 7.4). The final concentration of the compound was 10^−4^ mM. After 1 h at 37 °C, 1 mL of the reaction mixture was treated with 250 µL of the Griess reagent sulfanilamide (4 g), N-naphthyl ethylene diamine dihydrochloride (0.2 g), and 85% phosphoric acid (10 mL) in distilled water (final volume: 100 mL). Later, the mixture was kept for 10 min at room temperature and absorbance was measured at 540 nm; 10–80 nmol/mL sodium nitrite standard solutions were used for the calibration curve. The yield in nitrite was expressed as percentage (%) NO2- (mo/mol) ± SE. Detailed results are summarized in Table 1.

### 2.2. Smooth Muscle Relaxant Activity

All synthesized molecules were tested in vitro for their pulmonary vein relaxant activity as reported previously [39], which was brought from the local slaughterhouse in ice-cold Krebs–Henseleit solution. The smooth muscles were cut into spiral strips. These strips were mounted in 15 mL isolated organ baths, containing Krebs–Henseleit solution, mixed with 95% O_2_ and 5% CO_2_ at 37 °C. The strip was allowed to equilibrate for 2 h under a resting load of 2 gm. The relaxation of muscle strip was recorded for each drug using force transducer multichannel physiography (BIOPAC MP35 SYSTEM) using sildenafil as standard. Detailed results are summarized in Table 1.

### 2.3. Docking Studies

Docking analysis was carried out using biopredicta module of the V life MDS 4.3 [50]. We utilized the crystal structure of the PDE 5 (PDB ID 2H42) to perform the docking simulations, which was downloaded from www.rcsb.org. After downloading the protein structure, protein cleaning was carried out using V life engine where water molecules were removed, and native hydrogens were retained in the structure and it was minimized using MMFF 94 until it reached root mean square (rms) gradient energy of 0.001 kcal/molA°. Conformational searches on all the developed molecules were carried out and several numbers of conformations were generated [51]. All the generated conformations were docked in the crystal structure of the PDE V in order to find the binding interactions in the reference binding site of sildenafil. Detailed results are summarized in Table 2.

### 2.4. Antioxidant Activity:

The DPPH scavenging activity was performed using a solution of 0.1 mΜ DPPH in methanol. A total of 1 mL of DPPH solution was added to 3 mL of test samples having concentrations of 10, 20, and 30 μg/mL in methanol and kept in the dark. Thirty minutes later, the absorbance was measured at 517 nm. A blank was prepared without adding the compound. Ascorbic acid at concentrations of 10, 20, and 30μg/mL was used as standard. The lower the absorbance of the reaction mixture, the higher the free radical scavenging activity. The capability to scavenge the DPPH radical was calculated using the following equation (1).
(1)% radical scavenging actvity=Abs Control−Abs Sample/Abs Control x 100
where, % RSA is the radical scavenging activity; Abs control is the absorbance of DPPH radical + methanol; Abs test sample is the absorbance of DPPH radical + test samples.

## 3. Results and Discussion

Bis (2-chloroethyl) amine hydrochloride salt (1) was treated with benzyl bromide in the presence of potassium carbonate as base to obtain compound **2**. The compound **2** transformed to compound **3** using a strong base NaH (sodium hydride) with benzyl carbonitrile followed by basic hydrolysis to get compound **4**. The compound **6** (Appendix A) was obtained by the formation of an amide bond with methyl 2-amino acetate via acid chloride **5** obtained using thionyl chloride. The crucial cyclisation process was achieved under strong acidic conditions to afford spiro compound **7** (Appendix A). Further one pot debenzylation and Boc protection gave a stable compound **8**. Acid catalyzed Boc deprotection afforded the key scaffold **9** as HCl salt. The nucleophilic substitution of compound **9** with the corresponding bromo furoxan^7^ led to the production of final furoxan-coupled spiro products (**10a–j**) ranging from good to moderate yields as shown in Scheme 1.

The ability of the synthesized compounds to release NO was assessed by detecting nitrite, which was the principal final product of NO oxidative metabolism, using the Griess reaction. All compounds exhibit excellent NO release ability that ranged from 11.18% to 14.63% compared to methyl phenyl furoxan (7%). While, ortho nitro **10d** and methoxy **10g** substituted phenyl of furoxan derivatives showed relatively higher NO release than the corresponding meta and para substituted products.

The vascular smooth muscle relaxant activity of the novel synthesized compounds was carried out using the goat aorta. To our delight, compound **10d** (1.50 ± 0.7) with ortho nitro substitution on the phenyl of the furoxan ring displayed the highest activity and the potent PDE5 inhibitor compared to the standard sildenafil (1.43 ± 0.5). The methoxy derivatives **10g** (1.69 ± 0.6), **10b** (2.0 ± 0.7), and **10i** (2.90 ± 0.9) followed by Bormo derivatives **10f** (20 ± 0.1) and **10j** (20 ± 0.1) are summarized in Table 1. A direct correlation between the smooth muscle relaxant activity and the percentage (%) of vasodilatation was observed. The percentage of vasodilatation of the synthesized compounds showed that compound **10i** with meta methoxy substitution on the phenyl of the furoxan ring exhibited the highest vasodilation activity (60%) and the three derivatives **10b**, **10d**, and **10g** exhibited more than 50% activity as mentioned in Figure 1.

The antioxidant effect of the designed compounds was also evaluated as Sildenafil citrate has protective effects on oxidative stress by inhibiting free radical formation and by supporting antioxidant redox systems. In vitro antioxidant activity of synthesized compounds was studied based on the radical scavenging effect of the stable DPPH (2,2-diphenylpicrylhydrazyl) free radical using ascorbic acid as a reference standard. The synthesized compounds showed moderate to good antioxidant activity. 

The results are expressed at three different concentrations (10–30 μg/mL) in percentage (%) and summarized in Table 1. The results showed that compound **10i** (88.88%) showed the highest antioxidant activity compared with the reference compound ascorbic acid (92.58%). The remaining compounds exhibited moderate activity in the range of 70.11–77.88%.

For the docking analysis of the synthesized derivatives, a key factor was analyzed to identify their pharmacodynamic potential. Docking simulations were carried out in the binding site of the human PDE 5 which is a key enzyme involved in the vasodilation of smooth muscles. The efficiency of the docking protocol was also ascertained by the docking of sildenafil which is also a known vasodilator and standard utilized in biological activity. All the synthesized derivatives have binding energies in the range of -33.05 to -55.62 kcal/ mol, clearly suggesting the favorable interactive potential of the synthesized derivatives towards phosphodiesterase 5. The compound **10f** exhibited the strongest binding affinity with phosphodiesterase 5 followed by **10a**, **10b**, **10g**, **10i**, **10j**, **10e**, **10h**, **10c**, and **10d** (Table 2), in accordance with our bioactivity data. The predicted docking pose of the most active compound **10f** formed one hydrogen bond with GLN352 (1.2A^0^) and displayed strong aromatic interactions with PHE357 (5.2A^0^) and PHE110 (4.9A^0^). Derivative **10a** showed hydrogen bond interactions with GLN352 (2.1A^0^) and aromatic interactions with PHE357 (5.3A^0^), and PHE110 (5.4 A^0^). The synthesized derivative **10b** exhibited a hydrogen bond with GLN352 (2.5A^0^) and displayed aromatic interactions with PHE357 (3.7A^0^).

Derivative **10g** showed a hydrogen bond interaction with LEU369 (1.8 A^0^) and an aromatic interaction with PHE 322 (5.4A^0^). The docking results of all the derivatives are given in Table 2 and Figure 2 shows the interactions of compounds **10a**–**j** with PDE 5.

## 4. Conclusions

The spiro-isoquinolino piperidine skeleton was studied for its PDE 5 inhibition activity along with modelling studies. As expected, the skeleton displays good PDE-5 inhibitory activity and modelling studies provide structural insights for the observed binding. The observed results can be manipulated further to enhance the inhibition to be of therapeutic significance. Based on the present results, the possibility of NO and/or H2S donor hybrids with potential PDE5 inhibitor activity particularly NO–Sildenafil, H2S–Sildenafil, and NO + H2S–Sildenafil can be explored for future.

## Figures and Tables

**Scheme 1 biomedicines-08-00121-sch001:**
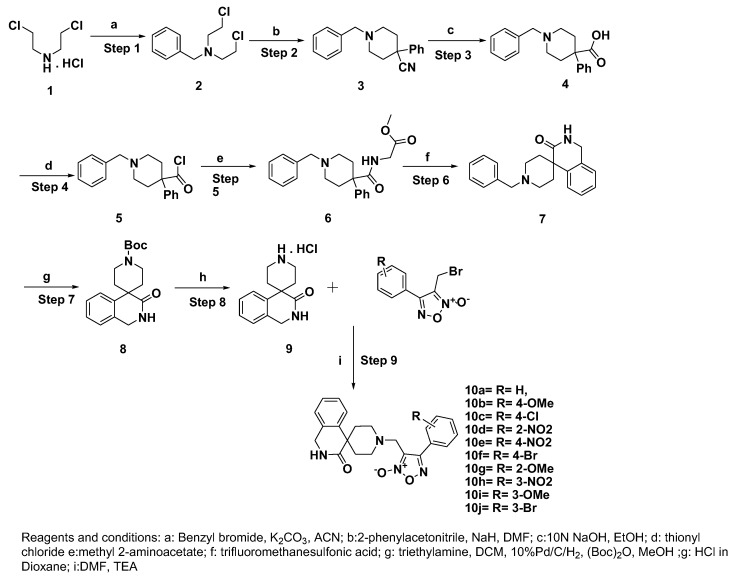
Synthesis of furoxan coupled spiro-isoquinolino piperidine derivatives.

**Figure 1 biomedicines-08-00121-f001:**
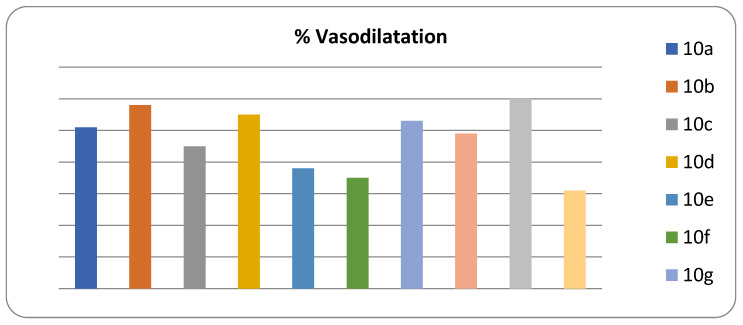
The % vasodilatation of **10a–j**.

**Figure 2 biomedicines-08-00121-f002:**
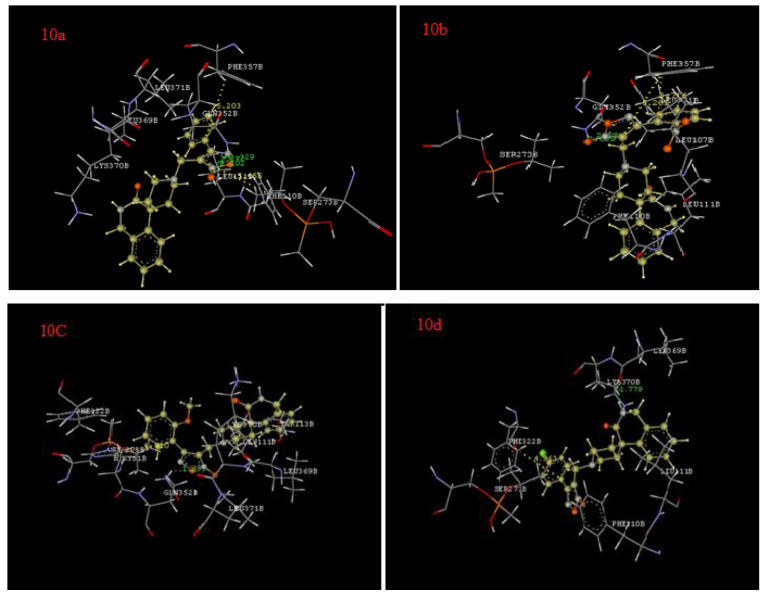
Interactions of **10a–j** with phosphodiesterase V.

**Table 1 biomedicines-08-00121-t001:** Biological activity results.

Compounds	NO Release	Smooth Muscle Relaxant Activity	Virtual Docking Score	% Antioxidant Activity		
	Assay % of NO_2_^–^ mol/mol + L-Cys 5 mM)	IC_50_	Phosphodiesterase V	10 μg/mL	20 µg/mL	30 μg/mL
**10a**	13.63 ± 0.46	59 ± 0.2	−42.044130	52.08	55.52	72.99
**10b**	14.27 ± 0.72	2.0 ± 0.7	−17.505997	52.88	65.78	79.14
**10c**	12.81 ± 0.21	45 ± 0.7	−42.382730	49.12	68.12	70.11
**10d**	14.63 ± 0.90	1.50 ± 0.7	−28.783650	55.11	65.82	77.88
**10e**	12.36 ± 0.31	38 ± 0.6	−16.155231	45.22	62.22	76.66
**10f**	11.18 ± 0.78	20 ± 0.1	−42.521134	48.98	67.52	69.22
**10g**	14.63 ± 0.88	1.69 ± 0.6	−48.472172	59.38	66.11	77.49
**10h**	13.41 ± 0.92	49 ± 0.5	−48.754056	55.77	65.98	71.18
**10i**	14.10 ± 0.54	2.90 ± 0.9	−70.164275	55.55	68.12	88.88
**10j**	11.18 ± 0.29	20 ± 0.1	−64.185966	54.36	63.21	73.44
**Methyl Phenyl Furoxan**	07.00 ± 3.00	-	-	-	-	-
**Sidenafil**	-	1.43 ± 0.5	-118.113503	-	-	-
**Ascorbic Acid**	-			83.83	85.79	92.58

**Table 2 biomedicines-08-00121-t002:** Docking results.

Compound	Docking Score	Key Interactions	
		Hydrogen Bond Interactions	Aromatic Interactions
**10a**	-47.593442	GLN352 2.1	PHE357 5.3 PHE110 5.4
10b	-46.984148	GLN352 2.5	PHE357 3.7
10c	-34.987223	GLN352 1.8	HIS351 4.1
10d	-33.052417	LEU369 1.7	PHE322 4.9
10e	-37.938047	GLN352 2.4	PHE110 4.9
10f	-55.627713	GLN352 1.2	PHE357 5.2 PHE110 4.9
10g	-42.682554	LEU369 1.8	PHE322 5.4
10h	-35.259930	LYS370 2.2GLN352 1.5	
10i	-42.193341	LYS370 2.5	PHE110 5.4
10j	-39.295003	GLN352 2.4	PHE110 5.4

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
