# Peer review of "Synthesis and Modeling Studies of Furoxan Coupled Spiro-Isoquinolino Piperidine Derivatives as NO Releasing PDE 5 Inhibitors"

_biomedicines, 2020, doi:10.3390/biomedicines8050121_

Round 1

Reviewer 1 Report

These are novel PDE5 inhibitors. A few comments:

First reading was confusing. It seems that you made errors in compound numbering. I see a series of compounds labeled 10a-10j. But sometimes a number 15 pops up ( see line 99 with 15i). Also in the chemistry section the compounds are numbered 15a-15j.

line 97: you write potential...... but probably you mean potent PDE5 inhibitors....

line 98: you write that the rest of the compounds are inactive but that certainly is not true for compounds with IC50 values between 2.9 and 59nM.  It would be good to extend the SAR discussion in relation to PDE5 inhibition.

Table 1 would improve if you add a column with the R substituent.

chemistry section

Add in all cases HRMS calculated and found and preferably in the same format. for 15a you only give HRMS with a number.

Author Response

Dear Reviewer, 

Thanks

Reviewer 2 Report

This manuscript “Synthesis and Modeling Studies of Furoxan Coupled Spiro-Isoquinolino Piperidine Derivatives as NO Releasing PDE 5 Inhibitors” describes the synthesis and characterization of NO-releasing PDE-5 inhibitors. The manuscript can be considered after thorough major revision.

  1. Please indicate what the numbers are in the abstract; eg. 10d (1.50+-0.6).
  2. In the manuscript, please clearly explain the design rational why the authors thought furoxan-coupled spiro products (10a-10j) ca be a PDE 5 inhibitor.
  3. Please show NO release kinetics (half-life, total release duration etc.) and graphes.
  4. Please briefly explain the experimental conditions how to measure vascular smooth muscle relaxant activity.
  5. Is the NO release only available in the presence of cystein? Please clearly describe and discuss the release mechanism and condition of NO from the product.
  6. Please discuss why the antioxidant activity was measured prior to showing the results.
    7. Overall, please revise the whole manuscript to show the purpose of the manuscript in more detail and clearly .

Author Response

Dear Reviewer,

Thanks

Reviewer 3 Report

This paper describes the  synthesis and modeling Studies of Furoxan Coupled Spiro-Isoquinolino Piperidine Derivatives as NO Releasing PDE 5 Inhibitors

The paper may have some potential interest but I have several point of concerns:

  1. The paper is written in poor English and this hinders the comprehension of the message. Extensive revision from a native English scientist is mandatory
  2. In the Abstract the Authors jump from NO to PDE inhibitors without any explained biological connection. They should try to be consequental and explain to the readers the biological background of their study and the links between NO and PDE inhibitors. 
  3. The Authors  have correctly compared their hybrid compounds to sidenafil as standard PDE5 inhibitors. They find that compounds 10d (1.50±0.6) and 23 10g (1.65±0.7) are moderate PDE 5 inhibitors as compared to Sidenafil (1.43±0.5). However, they are not better than Sidenafil. Why should these compounds be preferred to Sidenafil and studies and economic efforts invested to develop an early stage compound in the preclinical setting that is less effective than a marketed compound with known efficacy and safety? 
  4.  They could answer to 3 indicating that starting from this platform of compounds additional SAR studies may help identify compounds that are more powerful than Sidenafil. The Authors could use this study as proof of concept to say that Sidenafil hybridized to NO could perform better than parental Sidenafil.Along this line, the Authors should indicate that NO-hybridization of parental cdrugs such as aspirin, isoxazoline compounds (VGX-1027) and antiretroviral protease inhibitors has been largely studied and that NO hybridization has been found to often improve and eventually extend the pharmacological properties of parental compound in suppresion of cancer and immunoinflammatory responses. Actually this part is completely lacking and I would urge the Authors to describe it adequately for search of complete information.

  Song JM, Upadhyaya P, Kassie F.Nitric oxide-donating aspirin (NO-Aspirin) suppresses lung tumorigenesis in vitro and in vivo and these effects are associated with modulation of the EGFR signaling pathway. Carcinogenesis. 2018 Jul 3;39(7):911-920. doi: 10.1093/carcin/bgy049.   Maksimovic-Ivanic D et al. Anticancer properties of the novel nitric oxide-donating compound (S,R)-3-phenyl-4,5-dihydro-5-isoxazole acetic acid-nitric oxide in vitro and in vivo.Mol Cancer Ther. 2008 Mar;7(3):510-20. doi: 10.1158/1535-7163.MCT-07-2037.   Paskaš S et al., Senescence as a main mechanism of Ritonavir and Ritonavir-NO action against melanoma.Mol Carcinog. 2019 Aug;58(8):1362-1375. doi: 10.1002/mc.23020. Epub 2019 Apr 17.     Paskas S et al., Lopinavir-NO, a nitric oxide-releasing HIV protease inhibitor, suppresses the growth of melanoma cells in vitro and in vivo.Invest New Drugs. 2019 Oct;37(5):1014-1028. doi: 10.1007/s10637-019-00733-3. Epub 2019 Feb 1.   Basile MS et al., Anticancer and Differentiation Properties of the Nitric Oxide Derivative of Lopinavir in Human Glioblastoma Cells.Molecules. 2018 Sep 26;23(10). pii: E2463. doi: 10.3390/molecules23102463.   Maksimovic-Ivanic D et al., The NO-modified HIV protease inhibitor as a valuable drug for hematological malignancies: Role of p70S6K.Leuk Res. 2015 Oct;39(10):1088-95. doi: 10.1016/j.leukres.2015.06.013. Epub 2015 Jun 28.   Fagone P et al., Effects of NO-Hybridization on the Immunomodulatory Properties of the HIV Protease Inhibitors Lopinavir and Ritonavir.Basic Clin Pharmacol Toxicol. 2015 Nov;117(5):306-15. doi: 10.1111/bcpt.12414. Epub 2015 May 25.   Rothweiler F et al., Anticancer effects of the nitric oxide-modified saquinavir derivative saquinavir-NO against multidrug-resistant cancer cells.Neoplasia. 2010 Dec;12(12):1023-30.   Maksimovic-Ivanic D et al., HIV-protease inhibitors for the treatment of cancer: Repositioning HIV protease inhibitors while developing more potent NO-hybridized derivatives?Int J Cancer. 2017 Apr 15;140(8):1713-1726. doi: 10.1002/ijc.30529. Epub 2017 Jan 20.   5. The Authors should indicate that NO, along with H2S and carbon monoxide represents the endogenous gaseous system that is implicated in control of several biological responses,   Fagone P et al., Gasotransmitters and the immune system: Mode of action and novel therapeutic targets.Eur J Pharmacol. 2018 Sep 5;834:92-102. doi: 10.1016/j.ejphar.2018.07.026. Epub 2018 Jul 20.  
Lazarević M et al., The Hâ‚‚S Donor GYY4137 Stimulates Reactive Oxygen Species Generation in BV2 Cells While Suppressing the Secretion of TNF and Nitric Oxide.Molecules. 2018 Nov 14;23(11). pii: E2966. doi: 10.3390/molecules23112966.   It is important to notice that some biological effects of NO and H2S are synergistic, including the efects mediated by PDE4 inhbition on relaxation of pig and human bladder neck.  Ribeiro ASF et al., Powerful relaxation of phosphodiesterase type 4 inhibitor rolipram in the pig and human bladder neck.
J Sex Med. 2014 Apr;11(4):930-941. doi: 10.1111/jsm.12456. Epub 2014 Feb 12     Since doube hybrids with NO and H2S are being generated   Hu Q et al., Novel Angiogenic Activity and Molecular Mechanisms of ZYZ-803, a Slow-Releasing Hydrogen Sulfide-Nitric Oxide Hybrid Molecule.Antioxid Redox Signal. 2016 Sep 10;25(8):498-514. doi: 10.1089/ars.2015.6607. Epub 2016 Jun 29. , the hypothesis of generating doube NO and H2S hybrids for PDE5 inhibition should be hypothesized

Author Response

Dear Reviewer,

Thanks

Round 2

Reviewer 2 Report

The authors clarify all questions arisen from the reviewer, so recommended to be accepted. 

Author Response

Dear Reviewer,

Thanks for accepting the manuscript for publication. We appreciate your effeort to review our manuscript.

Regards

Reviewer 3 Report

The Authors have  not addresed my criticisms and what they did appears to me very superficial.

1.

In response 3, they admit that this is an exploratory paper that aims to identifying new and more potent molceuls. Why not to wait until then there ? Biomedicine is an authoritative journal

Note that they do nt even discuss this caveat among several limitations of the study. The SAR studies that I have indicated for future studies are not mentuioned as it is not mentioned the possibility to generate Sidenafil-NO or Sidenafil-NO+H2S

They should argument much better why Biomedicine should publish this paper as it stands and what perspectives it offers

2. 

In response to my point nr 4 where I have indicated some 8-9 references to quote and discuss and have also indicated the relevance to add an introductory concept on H2S and double NO and H2S donors they have answered with 4 lines and 2 papers quoted (none of them of thos I had suggested).

3. Furoxan may have genotoxic properties and hence they should consider to use different NO donors, such as NO covalently linked as it has been shown in several of the papers I have suggested

I find this a very unusual and approximative manner of conducting revision for peer reviewed journals such as Biomedicines. If the Authors are not familiar with usual guidelines of peer review process they should ask advice from Colleagues.

Author Response

Dear Reviewer

Thanks for your nice suggestions. We have addressed the questions raised by you. All the modified parts are marked in red.

The Authors have not addressed my criticisms and what they did appears to me very superficial.

  1. In response 3, they admit that this is an exploratory paper that aims to identifying new and more potent molceuls. Why not to wait until then there? Biomedicine is an authoritative journalNote that they do nt even discuss this caveat among several limitations of the study. The SAR studies that I have indicated for future studies are not mentuioned as it is not mentioned the possibility to generate Sidenafil-NO or Sidenafil-NO+H2S.They should argument much better why Biomedicine should publish this paper as it stands and what perspectives it offers

Present study is guiding torch of possibility of hybrid PDE5 inhibitor to treat male erectile dysfunction. In page 76-88 and Page 362-364 addressed criticisms. A manuscript usually prepared once a story get complete and we feel, the study provides a complete story of the mentioned things.

  1. In response to my point nr 4 where I have indicated some 8-9 references to quote and discuss and have also indicated the relevance to add an introductory concept on H2S and double NO and H2S donors they have answered with 4 lines and 2 papers quoted (none of them of thos I had suggested).

 The answer of four line was the summary with two review articles quoted. Although the present study mainly focused only on role of NO not other endogenous gaseous systems. However, based on suggestions concept of H2S discussed.

  1. Furoxan may have genotoxic properties and hence they should consider to use different NO donors, such as NO covalently linked as it has been shown in several of the papers, I have suggested

Unpublished preliminary results with low NO releasing compounds such as organic nitrates had no significant impact on activity. Hence furoxan moiety was preferred. Although it might show genotoxic effect, the toxicity can be reduced in the presence of oxyhaemoglobin.

I find this a very unusual and approximative manner of conducting revision for peer reviewed journals such as Biomedicines. If the Authors are not familiar with usual guidelines of peer review process they should ask advice from Colleagues.

Response

The manuscript is being reviewed by three potential reviewers. Other two reviewers have found the article suitable and the authors are well versed with procedures and guidelines of revision and peer review system.  We appreciate your criticism. Thanks

Round 3

Reviewer 3 Report

The Authors have satisfactorily addressed my criticisms